# Tunable, Low–Cost, Multi–Channel, Broadband Liquid Crystal Shutter for Fluorescence Imaging in Widefield Microscopy

**DOI:** 10.3390/mi13081310

**Published:** 2022-08-13

**Authors:** Yan Gong, Bo Li, Cheng-You Yao, Weiyang Yang, Qi Hua Fan, Zhen Qiu, Wen Li

**Affiliations:** 1Electrical and Computer Engineering Department, Michigan State University, 428 S Shaw Ln, #2120, East Lansing, MI 48824, USA; 2Biomedical Engineering Department, Michigan State University, 775 Woodlot Dr, East Lansing, MI 48824, USA; 3Chemical Engineering and Materials Science Department, Michigan State University, 428 S Shaw Ln, #2100, East Lansing, MI 48824, USA

**Keywords:** liquid crystal shutter, nematic liquid crystal, fluorescence imaging, tumor imaging, broadband

## Abstract

Bistable liquid crystal (LC) shutters have attracted much interest due to their low energy consumption and fast response time. In this paper, we demonstrate an electrically tunable/switchable biostable LC light shutter in biological optics through a three–step easy–assembly, inexpensive, multi–channel shutter. The liquid crystal exhibits tunable transparency (100% to 10% compared to the initial light intensity) under different voltages (0 V to 90 V), indicating its tunable potential. By using biomedical images, the response time, resolution, and light intensity changes of the LC under different voltages in three common fluorescence wavelengths are displayed intuitively. Particularly, the shutter’s performance in tumor images under the near–infrared band shows its application potential in biomedical imaging fields.

## 1. Introduction

Liquid crystals (LCs) have greatly promoted the development of many optical devices for various applications, such as LC displays [1,2], tunable Fresnel lenses [3,4], and time–resolved fluorescence microscopy [5]. Accurately controlling the alignment of LC is critical for advanced LC devices such as shutters, sensors, and modulators [6,7,8,9,10,11,12,13,14]. With the development of biological science, many new fields have emerged, such as optogenetics. As a unique optical instrument, LC has great potential to be used in these new fields. At present, the LC shutters available on the market are primarily made using polymer–dispersed LC, LC gel, cholesteric LC (ChLC), and dye–doped LC [15,16,17,18,19,20,21,22]. These LC shutters share a similar working principle; that is switching between two states, transparent and opaque, to adjust the intensity of incident light (on/off) [23]. Among different LCs, nematic liquid crystal (NLC) has unique elongated rod–shaped molecules which align parallel to a specific direction in space, resulting in its large inherent birefringence and dipolar reaction [24]. Therefore, the external electric field can affect NLC to produce an electro–optic response order of a magnitude higher than conventional dielectrics, e.g., LiNbO_3_ [24]. In addition, many of these LC devices require microfabrication techniques, such as chemical vapor deposition (CVD), sputtering, vacuum deposition, etc. [25,26], to deposit indium tin oxide (ITO) [27] as transparent conducting contacts. These microfabrication methods require special expertise, a cleanroom environment, and expensive equipment, which limits LC device usage to a certain extent. Therefore, a non–microfabrication method to construct LC devices can effectively reduce the overall cost of devices. Likewise, the shutter performance of LCs for biological imaging over a broad band requires further study, especially at near–infrared (NIR) wavelengths.

Under these circumstances, we designed and implemented a low–cost, mm–scaled, tunable NLC shutter by sandwiching NLC between indium tin oxide (ITO)–coated polyethylene terephthalate (PET) thin films. By accurately cutting the ITO layer and designing the control circuit, this NLC shutter shows the potential for multi–channel, individually addressable control. The characterization of the NLC and its potential application was also investigated in this study. In particular, the following five parameters were measured (under bright field and fluorescence test): the transparency of the NLC under different voltages, the light intensity distribution, the average passing light intensity, the optical fluorescent resolution, and the response time.

## 2. Materials and Methods

### 2.1. Sandwich–Structured NLC Shutter

A simple, rapid, and low–cost method was developed to fabricate disposable NLC shutters (Figure 1). This method can achieve small patterns of 400~500 µm without using microfabrication and cleanroom facilities. As illustrated in Figure 1A, a commercially available ITO–coated PET (Sigma–Aldrich, St. Louis, MO, USA) was utilized as a substrate. The ITO–PET substrate was chosen mainly due to its low cost and easy patterning process. To create multichannel ITO contacts, a computer–aided craft cutter (Silhouette Cameo 1 Cutting Tool, Silhouette America^®^, Inc., Lindon, UT, USA) was used to pattern the ITO electrode grids on the PET substrate. As shown in Figure 1A, the cutting damage was very localized and limited to 20 µm around the cutting areas. To further validate the integrity of the ITO film, the sheet resistance of the ITO film around the cutting areas was measured before and after cutting, and the average sheet resistance was 310.25 Ω/cm^2^ and 321.75 Ω/cm^2^, respectively, with no significant change. A 100–nm–thick SiO_2_ layer was sputtered (Model PRO Line PVD 75 Thin Film Deposition System Platform, Kurt J. Lesker, Jefferson Hills, PA, USA) as an insulating coating on the patterned electrodes to reduce ITO damage under high voltage and high humidity conditions. A fluidic chamber was built using a 114–um–thick double–sided tape (3M™ 9474LE). The NLC (N–(4–Methoxybenzylidene)–4–butylaniline) was purchased from MilliporeSigma (MilliporeSigma, Burlington, MA, USA) and used without further mixing. The operating temperature of the NLC is usually ranged between 21 °C and 47 °C [28], the birefringence (Δn) and viscosity coefficient at room temperature are 0.15 [29] and 0.21 Pa·s [30], respectively. The NLC was over–dripped on the bottom panel, and the excess NLC was squeezed out by the extrusion of the upper panel to avoid air bubbles. After loading the NLC into the chamber, another patterned ITO electrode array was aligned and bonded to the bottom electrode array to enclose the NLC inside the chamber. Then the chamber was sealed with polydimethylsiloxane (PDMS, SYLGARD™ 186), which isolates the NLC from oxygen and moisture to prevent material degradation.

### 2.2. Fluorescence Phantom and Tissue Preparation

To characterize the broad band performance of the NLC, three phantoms were prepared with agar to mimic optical properties typically observed in biological tissues. Low melting point agarose (1%, Sigma) was prepared with 0.2 µm diameter carboxylate–modified microspheres beads. Three different kinds of beads were used in the fluorescent tests: 505/515 nm yellow–green fluorescence beads (Invitrogen F8787, Thermo Fisher Scientific, Waltham, MA, USA), 540/560 nm orange fluorescent beads (Invitrogen F8809, Thermo Fisher Scientific, Waltham, MA, USA), 660/680 nm dark red fluorescent beads (Invitrogen F8789, Thermo Fisher Scientific, Waltham, MA, USA). The concentration of the three kinds of beads was 5 μL/mL, which contains 4.5 × 10^12^ beads/mL. The 1% agarose mixed with deionized water was heated to 100 °C. After cooling to ~60 °C, beads were added to the agarose gel to make the fluorescence phantom. The phantom was then mixed by a vortex mixer (Vortex–Genie 2, Scientific Industries, Bohemia, NY, USA) for one minute to ensure the beads were evenly distributed in agarose. The phantom was poured onto a glass slide for further use. The camera was set to a sequential recording mode. Within a 600 s period, the camera took one photo every 100 ms, and approximately 6000 photos were taken in total. During this period, the NLC shutter was operated 6 times, and the response time of the NLC was calculated from this series of images.

For ex vivo tissue imaging, all the animal experiments were performed in accordance with the guidelines approved by the Institutional Animal Care & Use Committee at the Michigan State University (IACUC, Protocol #AUF 06/18–082–00). The tumor tissue was harvested from a 4–month–old female mouse (MMTV/PyMT) and topically stained with 100 µg/mL indocyanine green (ICG) NIR fluorescent dye for 10 min, then the tissue was rinsed 5 times with phosphate–buffered saline (0.01 M PBS, 7.4 pH) for 5 min each. The sample tissue was then placed on a clean glass slide and mounted on the focal plane for imaging.

### 2.3. Experiment Setup

The assembled NLC shutter was characterized using a wide–field microscope (WFM). As shown in Figure 2A, the laser light source was placed above the test objects and spread the light equally on the NLC surface. All incident light first passed through the NLC device was projected onto the target object, and then imaged by a camera. For bright field testing, a 1951 United States Air Force (USAF) resolution test chart target was used as the target object. The incident light was projected by a fiber–lite illuminator (Dolan–Jenner Fiber–Lite High–Intensity Illuminator 180 Series, US), and the entire microscope was surrounded by opaque materials to ensure that most of the light entered directly above 90 degrees to the NLC device. For the fluorescence test, fluorescent beads were used as target objects. The incident light was provided by lasers with different wavelengths. The entire microscope was surrounded by opaque materials to ensure complete darkness during experiments. Due to the self–luminous properties of fluorescent beads, light was emitted from under the NLC device, passed through the NLC, and then reached the microscope camera at the top.

To record the switching process of the NLC, a control circuit was designed to synchronize the switch timing of the NLC shutter and camera. As shown in Figure 2C, the input power (HP^®^ E3612A) of the NLC shutter was switched by an optocoupler (PC817X series 4 pin general purpose photocoupler), and this optocoupler was also controlled by a function generator (Siglent SDG2042X Function/Arbitrary Waveform Generator) connected to the camera. The function generator outputted a 5 V square waveform for 5 s in every 100 s. The camera started recording when it received an external signal. Simultaneously, the optocoupler turned on the NLC input power, the voltage was applied to the NLC shutter, and the shutter started to operate.

Two types of tests were carried out, which were: (1) the bright field test focused on characterizing the performance of the NLC in white light illumination under different voltages and measuring its response time, light intensity changes, and resolution; and (2) the fluorescence test focused on measuring the response time, light intensity changes, and resolution at a specific wavelength under different voltages.

### 2.4. Data Acquisition and Analysis

The images captured by the camera were processed by MATLAB (MATLAB 2021b, The MathWorks, Natick, MA, USA) for intensity normalization, light intensity distribution, response time, and Gaussian fitting. As shown in Figure 3, the response time (or fall time) is defined as the time required for a normalized light intensity to change from a specified high value to a specified low value. In this study, these values were 90% and 10% of the step height (the change from the high state level to the low state level). The resolution of the bright field test was calculated using a knife–edge method. The original knife–edge image was obtained by processing the USAF target image covered by the NLC. The edge spread function (ESF) was obtained from the original knife–edge image. The line spread function (LSF) was obtained by taking the derivative of the ESF (where the absolute values were normalized to have a unit maximum). The Gaussian function was used for noise reduction to fit the original LSF. Finally, the full width at half maximum (FWHM) was estimated by LSF. The resolution calculation of the fluorescence test was calculated using the point spread function (PSF). The PSF was obtained from the original fluorescence image. The Gaussian function was used to fit the PSF curve for noise reduction. The FWHM was estimated from the Gaussian curve.

## 3. Results and Discussion

### 3.1. General Overview of NLC

The images captured by the camera were processed by MATLAB for intensity normalization. Figure 4A shows the transmittance of the NLC shutter under different voltages and wavelengths, measured by a spectrometer (Ocean optics flame–miniature fiber optic spectrometer). Figure 4A shows that the longer the wavelength, the larger the on/off ratio of the NLC achieved, in other words, the better the shutter performance. For example, in the 500 nm wavelength, when the NLC is between 0 V and 75 V, the difference in its transmittance is about 30%. At 1400 nm, the difference in transmittance of the NLC expanded by 50% between 0 V and 75 V. These results demonstrate that the NLC can work within a broad band range of 500 nm to 1600 nm, and the application potential of the NLC in the NIR band is good. Among different tested voltages, the NLC barely changed at 5 V. In the range of 10 V to 75 V, each increase in voltage will bring a significant change in transmittance. When the applied voltage on the NLC was above 75 V, the change caused by the increase in voltage decreased, and the transmittance was stabilized at 2~3%. Figure 4B shows the effect of different voltages on the response time. The influence of the activation voltage on the NLC was multi–faceted. Taking the fluorescence image at the 560 nm wavelength, the response time was about 500 ms at 75 V. At 45 V, the reaction time was extended to 1 s. When the NLC was activated, the average light intensity attenuated more as the voltage increased, and the NLC responded faster, accordingly.

### 3.2. Bright Field Test

To measure the NLC shutter response time in bright field, a 1951 USAF resolution test chart was used as the imaging target. Through the continuous on and off cycling of the shutter in a white light environment, the response time was determined by sampling the changing area. For the bright field test, the bright area at 0 s was selected and set as the starting point (initial light intensity value). The light intensity changed in this region for the first 1 s and was then averaged for the next six cycles. Finally, the time taken to change the average light intensity from 90% to 10% of the step height was calculated as the response time (Figure 5E). At 15 V and below, the light intensity change was insignificant, therefore these sets of data were not used in the calculation. The performance of the NLC at different voltages in the bright field is shown in Figure 5D. The effect of the activation voltage on the NLC is reflected in the change of light intensity when the NLC was fully closed. However, because of the slow relaxation of the NLC [14,15], the turn–off recovery time of the NLC was approximately 3 s, which was much slower than its turn–on time. As shown in Figure 5E, under 75 V, the turn–on response time of the NLC was relatively rapid. As such, it took approximately 500 ms to reduce the light intensity from 90% to 10% of the normalized light intensity step height. Figure 5A shows the image changes before and after the NLC shutter was opened and closed at 75 V. By comparing the light intensity distribution, the overall brightness of the masked image decreased after the NLC was activated. In other words, the brightest part of the original image may still be the brightest area of the masked (where the NLC covered) image, but the difference in light intensity between the masked area and the surrounding area was significantly reduced. Therefore, the image cannot be observed macroscopically, and the purpose of shuttering was achieved. In Figure 5B, with the NCL off (at 0 V), the bright field resolution was characterized by the ESF. In Figure 5C, the LSF was obtained by taking the derivative of the ESF. The Gaussian function was used to fit the original LSF and the estimated FWHM was 8.43 µm.

### 3.3. Fluorescence Test

The fluorescent performance of the NLC shutter was characterized by the response time, resolution, and light intensity changes under different voltages with 515 nm (25 mW power, 10 ms exposure time), 560 nm (25 mW power, 10 ms exposure time), and 680 nm (40 mW power, 10 ms exposure time) fluorescent beads, respectively. As shown in Figure 6, Figure 7 and Figure 8, the NLC performed roughly the same at different wavelengths. Generally speaking, the NLC performs well at different wavelengths, and can effectively block light by reducing 90% of initial light intensity within a wavelength range of 515 nm (green) to 680 nm (red). However, there are still differences worth exploring in detail. Comparing Figure 6C, Figure 7C and Figure 8C, the longer the wavelength of the incident light, the better the shuttering effect of the NLC under the same voltage, which is consistent with that shown in Figure 4A. The reduction in light intensity increased from around 80% (Figure 6C and Figure 7C) to more than 90% (Figure 8C). However, as the wavelength increased, the response time increased slightly. More importantly, when comparing Figure 7D and Figure 8D, the curve obtained at 680 nm was smooth, and no fluctuation was observed after the intensity reached the lowest level. The comprehensive performance of the NLC device at the wavelength of 680 nm was better than that at the wavelength of 560 nm. 

In Figure 6B, Figure 7B and Figure 8B, the PSF under the three wavelengths (515 nm, 560 nm, 680 nm) were estimated and fitted by Gaussian curves, and the corresponding FWHM values were 6.34 µm, 6.34 µm, and 10.4 µm. Since the size of the beads is 220 nm, which is much smaller than the minimum resolution of the camera (1.7 µm), one bead can act as a pixel that is emitting light (a point source). The PSF was calculated from the image containing a single pixel emitting point. Technically, the smaller the FWHM, the smaller the feature it can display and therefore the higher the imaging resolution. Further analysis of the image shows that, like the results in the bright field imaging, the NLC under the fluorescence imaging also exhibited a masking effect, resulting in the reduced difference in light intensity between the peak and the surrounding areas to achieve the purpose of shuttering. Unlike the bright field test, because the wavelength of light in the fluorescence test was relatively consistent, the shuttering effect of the NLC varied under different wavelengths. At 515 nm wavelength (Figure 6A), although the shuttering effect was observed, the peaks from different beads were visible and distinguishable after the NLC was activated, as shown in the distribution of light intensity under 75 V. Under the same voltage conditions, the NLC completely suppressed the peaks at both 560 nm (Figure 7A) and 680 nm (Figure 8A). This indicates that the NLC has considerable potential to be used in bioimaging applications with yellow fluorescent protein (YFP, 560 nm) and red fluorescent protein (RFP, 680 nm).

### 3.4. Tissue and NIR Testing

After a series of preparations, freshly harvested tumor tissues were stained with 100 µg/mL ICG and placed on slides ready for imaging. For the fluorescence imaging, an output of 3.5 mW, 808 nm laser beam with a beam size of 5 mm in diameter was distributed on the tissue surface. A dual–channel NLC shutter completely covered the tumor area. The collected fluorescent light was passed through a 1064 nm–long pass filter (1064 nm EdgeBasic™ best–value long–pass edge filter, SEMROCK, Rochester, NY, USA) for NIR imaging. When the NLC was activated, only the top channel shutter was closed, and the bottom channel remained open. For both channels, the fluorescent signals were collected and compared to each other when the shutter was on and off. Representative images from NLC turn on and turn off conditions with the same magnification of the same tissue are shown in Figure 9A,B. For the NLC–off image, the structure of the tissue surface can be relatively clearly distinguished. Considerable detail was captured and can be observed. In contrast, when the top channel of the NLC device was on, the top area of the tissue disappeared from the image due to the shuttering effect. Although some afterimages remain, the image of the shuttered region cannot be recognized, and the outline of the tumor was blurred. This implies that the received light intensity had been reduced to a relatively low level. The light intensity map shows that the NLC effectively reduced the overall light intensity while eliminating the difference between the image area and the surrounding area.

### 3.5. Comparison of Shutters 

As shown in Table 1, although other reported LC shutters had different LC formulations and different application scenarios, their core mechanisms were the same. The NLC devices presented in this study have many unique advantages. First, the NLC device demonstrated the shuttering capability in a relatively wider bandwidth of 515–1100 nm, not only in the visible light band, but also in the NIR band. Moreover, in the visible band, the shuttering capability of our NLC device was demonstrated under bright field and fluorescence imaging, and its characteristics were quantified in terms of the on/off intensity map, resolution, light intensity changes, and average response time. Second, although the NLC device has not been optimized, its on/off ratio is comparable to that of other devices. Finally, our NLC device featured low–cost, rapid fabrication and disposable characteristics. This allows the NLC shutter to be quickly integrated with other experiments without special expertise and equipment.

## 4. Conclusions

In this study, the properties of the NLC under bright field and fluorescence imaging were characterized. Further, the shutter performance of the NLC device on tumor tissues at NIR wavelengths was also demonstrated. The data shows that such disposable, fast, and low–cost fabricated NLC shutters can perform well over a broad band (515–1100 nm). More specifically, as the wavelength increases from 515 to 680 nm, the NLC performance can be further improved with minimum light intensity decreasing from 20% to <10%. These characteristics can be used in a variety of bio–optical scenarios. Compared to commonly used mechanical shutters, the NLC shutter has no moving parts, because the NLC shutter controls the refractive index of LC molecules by voltage to realize the phase retardation of light. In the future, further optimization of the NLC shutter to increase the resolution and response speed will be investigated. The activation voltage (up to 75 V) of the current LC shutter was relatively high, which can be further improved in the future by reducing the overall thickness of the fluidic chamber and trying different LC materials. Converting the electrical drive input from direct current (DC) to alternating current (AC) can also effectively reduce the activation voltage by 50% under the same transmittance [35]. We expect that the introduced NLC material and shutter design can be applied to wide–field microscopy and bio–optical fields as well as many other applications.

## Figures and Tables

**Figure 1 micromachines-13-01310-f001:**
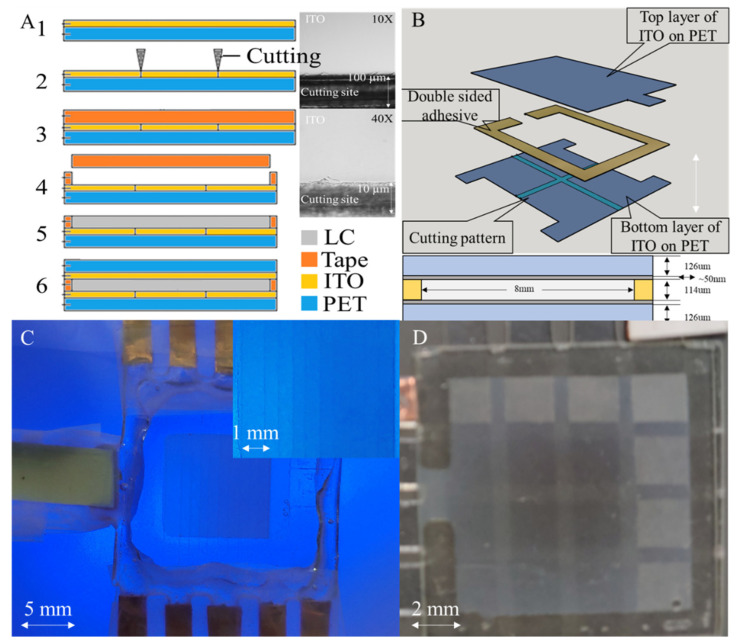
(**A**). Schematic of the device fabrication process and 10X and 40X optical microscopic image of the cutting site. (**B**). Schematic diagram showing the layer structure of the NLC shutter. (**C**). A working NLC shutter under different voltages (10 V to 100 V from left to right, with 10 V increment per channel), demonstrating its tuning capability. (**D**). A working NLC shutter showing multiple channel controllability (4 mm^2^ per channel).

**Figure 2 micromachines-13-01310-f002:**
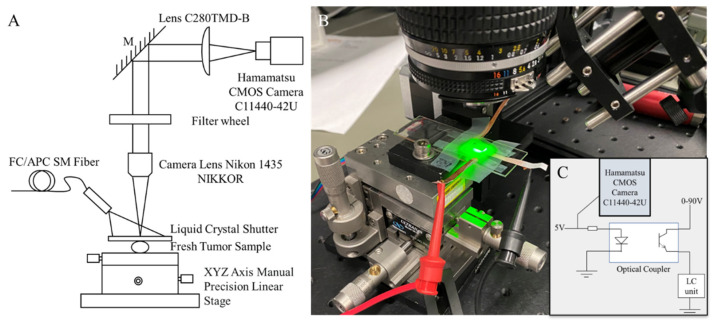
(**A**). Schematic of the WFM setup. (**B**). An image of the actual experimental setup. (**C**). Schematic diagram of the control circuit.

**Figure 3 micromachines-13-01310-f003:**
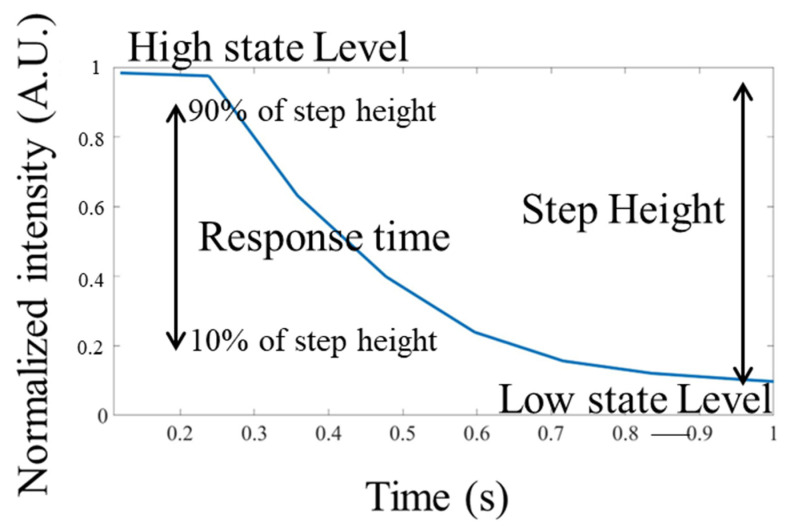
Schematic diagram for calculation of response time.

**Figure 4 micromachines-13-01310-f004:**
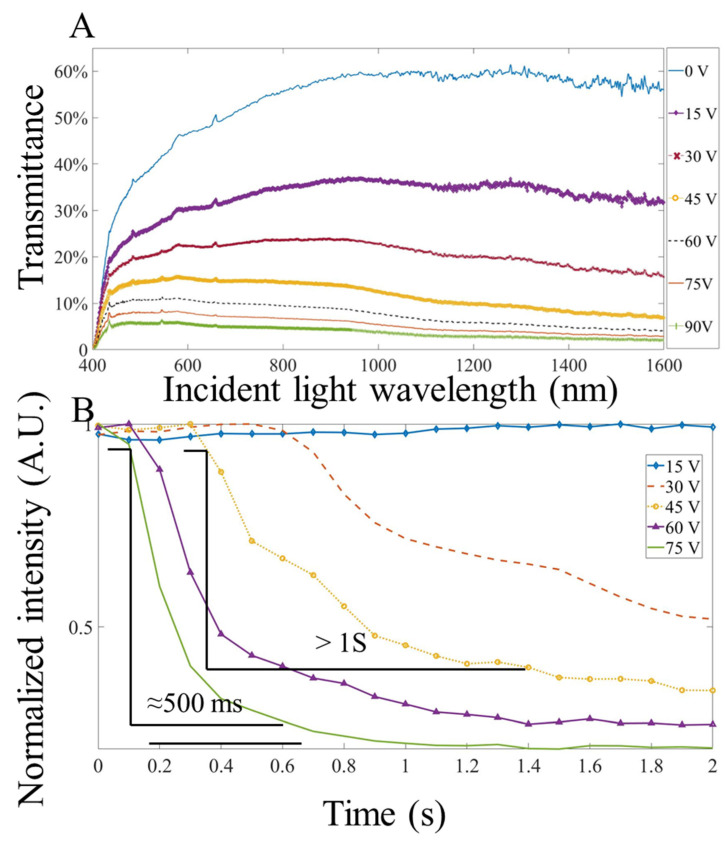
(**A**). The transmittance of the NLC at different wavelengths from 0 to 90 V. (**B**). Response time at different activation voltage (estimated at 560 nm wavelength).

**Figure 5 micromachines-13-01310-f005:**
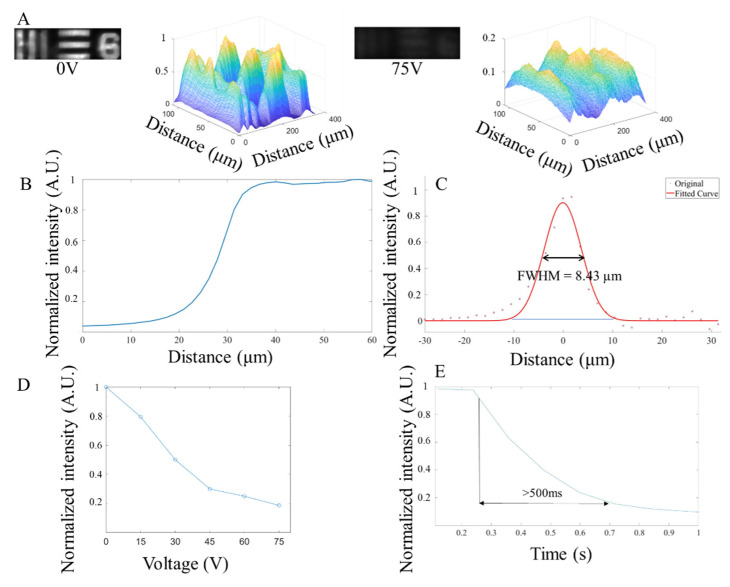
The NLC performance of the liquid crystal under the bright field condition. (**A**). The microscope image through the NLC (1951 USAF resolution test chart) and the light intensity distribution color map at 0 V (**left**) and 75 V (**right**). (**B**). The ESF of the NLC. (**C**). Estimation of the NLC’s LSF and Gaussian fitting curve at 0 V. (**D**). Average measurement of light intensity changes under different voltages. (**E**). Response time (change from 90% to 10% of the normalized light intensity step height) of the NLC under 75 V.

**Figure 6 micromachines-13-01310-f006:**
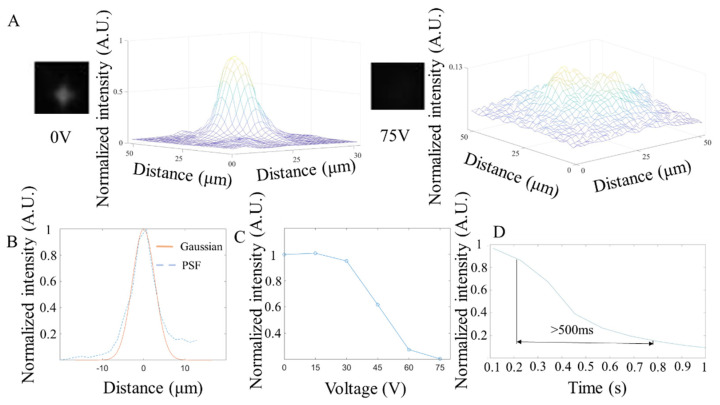
Characterization of the NLC at 515 nm wavelength. (**A**). The microscope image of the NLC and the light intensity distribution color map at 0 V (**left**) and 75 V (**right**). (**B**). The NLC’s point spread function and Gaussian fitting curve at 0 V. (**C**). Average measurement of light intensity changes under different voltages. (**D**). Average response time (change from 90% to 10% of the normalized light intensity step height) of the NLC under 75 V (N = 6).

**Figure 7 micromachines-13-01310-f007:**
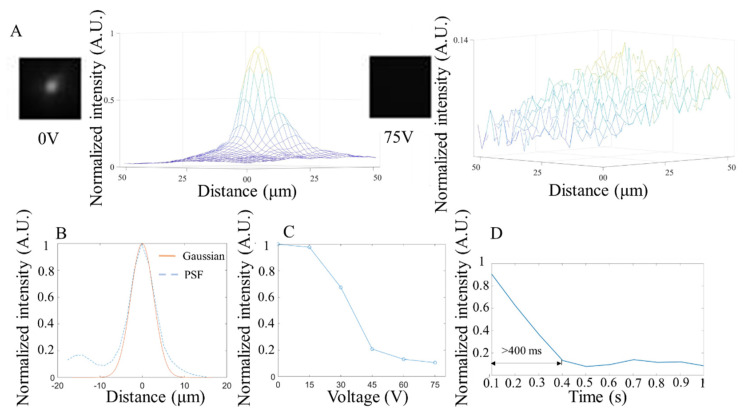
Characterization of NLC at 560 nm wavelength. (**A**). The microscope image of the NLC and the light intensity distribution color map at 0 V (**left**) and 75 V (**right**). (**B**). The NLC’s point spread function and Gaussian fitting curve at 0 V. (**C**). Average measurement of light intensity changes under different voltages. (**D**). Average response time (change from 90% to 10% of normalized light intensity step height) of the NLC under 75 V (n = 5).

**Figure 8 micromachines-13-01310-f008:**
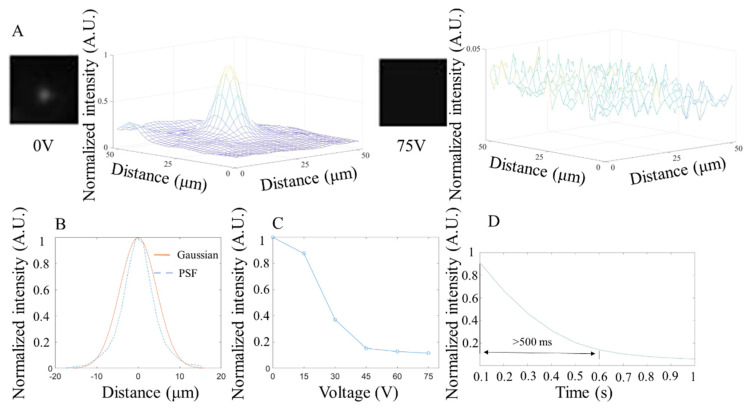
Characterization of NLC at 680 nm wavelength. (**A**). The microscope image of the NLC and the light intensity distribution color map at 0 V (**left**) and 75 V (**right**). (**B**). The NLC’s point spread function and Gaussian fitting curve at 0 V. (**C**). Average measurement of light intensity changes under different voltages. (**D**). Average response time (change from 90% to 10% of the normalized light intensity step height) of the NLC under 75 V (n = 5).

**Figure 9 micromachines-13-01310-f009:**
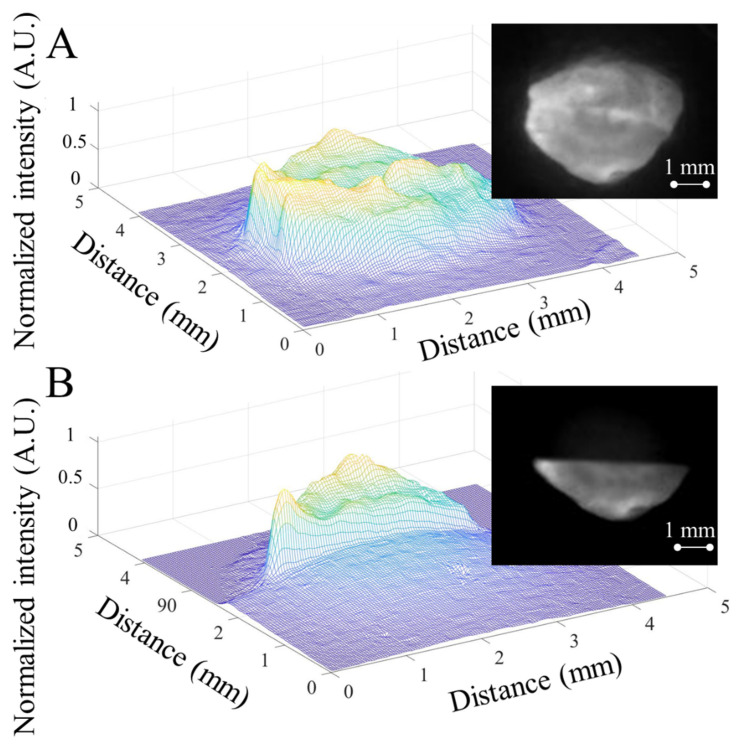
(**A**). The actual image and light intensity distribution graph of tumor tissue fully covered by the dual–––channel NLC shutter (with both channels deactivated) under 1064 nm ICG NIR. (**B**). The actual image and light intensity distribution graph of tumor tissue fully covered by the dual––channel NLC shutter (with top channel activated) under 1064 nm ICG NIR.

**Table 1 micromachines-13-01310-t001:** Summary of the NLC shutter performance as compared with for other reported LC shutters.

LC Type	Working Range	Deactivated Transmittance (%)	Activated Transmittance (%)	Reference
NLC (This article)	515 nm to 1100 nm	>60%	>5%	NA
Dye–doped LC	590 nm	50.2%	5.1%	[31]
Cholesteric (Ch) LC	750 nm to 1120 nm	>60%	35% to 1.04%	[32]
Dye–doped ChLC	1000 nm/2000 nm	0.795%/1.94%	72.7%/34.7%	[33]
polymer–networked LC	400 nm to 650 nm	65.5%	2.3%	[23]
Dye–doped NLC	400 nm to 650 nm	84% (35 wt%)	10% (35 wt%)	[34]

## Data Availability

Not applicable.

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
