# Peer review of "Tunable, Low–Cost, Multi–Channel, Broadband Liquid Crystal Shutter for Fluorescence Imaging in Widefield Microscopy"

_micromachines, 2022, doi:10.3390/mi13081310_

Round 1
Reviewer 1 Report
The authors propose a kind of nematic liquid crystal shutter. The English level of writing is good, and the experimental data provided are detailed. In my opinion, this manuscript meets journal standards, provided that some issues are addressed to further improve the manuscript.
1. In the abstract, the authors mentioned “Although many transparent conductors have been developed and widely used, the high cost due to the complex fabrication method required by LC devices constricts the usage of these devices”, can the author explain this situation further so that readers can have a comprehensive understanding.
2. How did the authors obtain the liquid crystal materials used, whether they were synthesized in the laboratory or provided by others?
3. Various parameters of the liquid crystal material used have not been provided in the manuscript, such as birefringence, viscosity coefficient, clearing temperature, and operating temperature range. These are the key indicators that directly determine the performance of the device and affect the application prospect of the device.
4. It seems that the liquid crystal inside the device is injected at room temperature, whether there is the influence of residual stress.
5. Why the authors chose ITO-coated PET as the substrates, in the band range of the experiments carried out, it seems that glass with more comprehensive stability is more suitable. Is it just about cost?
6. The thickness of the SiO2 needs to be provided.
7. Due to the fragility and lack of flexibility of the ITO film, the use of a cutting machine may cause unavoidable damage. The authors need to clarify this and provide optical microscopic images of the cut ITO layers.
8. In my understanding, using a 170-μm-thick tape means that, ideally, the liquid crystal layers from which the device is made share the same thickness. Why in Figure 1b, this thickness appears to be 114μm.
9. Whether it's 170µm or 114µm, why choose this thickness. In other words, how thicknesses below or above this value will affect the performance of the device.
10. The voltage range used is too high (up to 75v), which indicates the need for an additional electrical drive input. What means can be used to reduce this scope, and to what extent?\
11. It seems that in the system, the collaborator's name is mistaken, if I guessed correctly, the correct one should be Qi Hua Fan.
Author Response
Thank you very much for your valuable comment. We read them and responded to them carefully. We have learned and improved a lot with your suggestions. Details are now in the attached word file.

Reviewer 2 Report
The paper described a low-cost, mm-scaled, tunable NLC shutter indium tin oxide (ITO)-coated polyethylene terephthalate (PET) thin films. Unfortunately, this idea is not that novel since researchers already reported the NLC shutter for projection display more than twenty years ago (Sergev et al. 1996). Before that, the application of the LC shutter in fluorescence imaging is also imported (Verwoerd et al, 1994). I am wondering that why the authors don’t cite these classic papers. Although the idea is not that novel, the NLC shutter presented by the authors have many advantages compared with other similar devices. For example, it is relatively low-cost, broadband and easy-fabricated. The authors also testified the performance and applications of the NLC shutter, such as brightfield, fluorescence and NIR imaging. Therefore, the work presented is interesting and deserves published in Micromachines.
I would still recommend the authors to address the following points:
(1) The authors claim that the principle of the current NLC shutter is different from other shutters. However, it is not clearly described in the context.
(2) Could the authors describe the background clearly and carefully in order to attract more attentions from different fields. The background is not enough emphasized in current manuscript. For example, the authors claim that the transparent conductive materials are also important, but they neither showing the reason nor providing literatures.
(3) The font in Fig.1A is blurry. Could the authors use thick font instead to improve the resolution of the image?
(4) The emergence order of figures is messy. For example, Fig.5 and Fig.6 are described in front of Fig.4, as seen from line 130 to line 135.
(5) It is very hard to differentiate these different curves only using colors. Maybe it is better to combine the symbols and line-shapes, such as dashed lines with circle, etc.
Author Response
Thank you very much for your valuable comment. We read them carefully and responded to them carefully. We have learned and improved a lot with your suggestions. Please see the attachment.

Round 2
Reviewer 1 Report
I recommand that the revised version be accepted directly.